# Clinical Evaluation of the VirClia IgM/IgG Chemiluminescence Tests for the Diagnosis of Tick-Borne Encephalitis in an Endemic Part of Norway

**DOI:** 10.3390/v16091505

**Published:** 2024-09-23

**Authors:** Åshild Marvik, Susanne Gjeruldsen Dudman

**Affiliations:** 1Department of Microbiology, Vestfold Hospital Trust, 3103 Tønsberg, Norway; aamarv@siv.no; 2Institute of Clinical Medicine, University of Oslo, 0313 Oslo, Norway; 3Department of Microbiology, Oslo University Hospital, 0372 Oslo, Norway

**Keywords:** tick-borne encephalitis, serology, tick, flavivirus, encephalitis, VirClia

## Abstract

The aim of this study was to evaluate the clinical usefulness of VirClia IgM/IgG single-assay chemiluminescence tests for the diagnosis of tick-borne encephalitis (TBE) in an endemic part of Norway. Patients hospitalized at Vestfold or Telemark Hospitals with suspected infection in the central nervous system (CNS) in the period between May 2021 and December 2023 were included, with 85 TBE cases identified. The VirClia IgM assay was positive in the initial serum sample in 75/85 cases, giving a sensitivity of 88.2% (95% CI, 79.4–94.2). The ReaScan TBE IgM rapid test was positive in 80/85 cases, with an estimated sensitivity of 94.1% (95% CI, 86.8–98.1). Vaccine breakthrough infections were the predominant cause of non-reactive IgM cases. The calculated specificity for the VirClia IgM was 95.8% (95% CI, 92.5–98.0). In conclusion, the sensitivity of the VirClia IgM was non-inferior to the ReaScan TBE IgM rapid test. However, isolated IgM reactive results must be interpreted with caution, since false-reactive results occur.

## 1. Introduction

Tick-borne encephalitis (TBE) constitutes a major health concern in Europe, the Russian Federation and northern Asia [1]. More than 3600 TBE cases were reported in Europe in 2022 [2]. The causative agent, the TBE virus (TBEV), is a flavivirus with five subtypes named according to the different geographical distributions, of which the European (TBEV-Eu) subtype is endemic in Norway [3].

Infectious tick bites are the commonest source of TBE, and *Ixodes ricinus* is the main vector for TBEV-Eu. TBEV infections can be asymptomatic, present as a transient febrile illness or lead to infection in the central nervous system (CNS), resulting in meningitis, encephalitis or meningoencephalomyelitis. Patients with CNS involvement most often experience a characteristic biphasic course [1,2]. After the initial viremic phase, characterized by fever and influenza-like symptoms, a symptom-free interval of approximately one week follows before the recurrence of high-grade fever combined with symptoms of CNS inflammation. However, in a recent publication from The Norwegian Tick-Borne Encephalitis Study (NOTES), 46% of 153 TBE cases had a monophasic course with no symptom-free interval reported [4].

In Norway, TBE is mandatory notifiable to the Norwegian Surveillance System for Communicable Diseases (MSIS). Until 2015, the incidence of TBE fluctuated, but thereafter, Norway saw a gradual increase in TBE cases, likely due to a combination of factors increasing *I. ricinus* distribution and human exposure. Climate changes, forest regrowth due to, e.g., a decreasing number of farms and people spending more time outdoors, especially during the COVID-19 pandemic, are probably all contributing factors [5,6]. A peak year was observed in 2023 with 94 domestic cases, most of them infected in Vestfold and Telemark counties [7].

TBE IgM antibodies are absent in the viremic phase, but the majority of patients will have detectable TBE IgM and IgG antibodies in serum at the time when CNS symptoms occur [1,8,9]. However, in immunocompromised patients and vaccine breakthrough infections (VBIs), IgM production can be haltered, leading to diagnostic difficulties [4,10,11,12]. Enzyme-linked immunosorbent assay (ELISA) is the main serological method of the laboratory confirmation of TBE [1]. Recently, automated single-assay chemiluminescence tests, the TICK-BORNE ENCEPHALITITS VIRCLIA IgM/IgG MONOTESTs, have also become available for the detection of TBE antibodies in serum/plasma. In addition, an immunochromatographic lateral flow assay, the ReaScan TBE IgM rapid test, has shown excellent performance [13].

The primary aims of this study were to evaluate the diagnostic accuracy of VirClia IgM/IgG for the diagnosis of TBE in an endemic part of Norway and to compare the VirClia IgM with the ReaScan IgM.

## 2. Materials and Methods

### 2.1. Case Data and Definitions

All 508 patients admitted to Vestfold or Telemark Hospitals in the period May 2021–December 2023 subjected to a VirClia IgM test were retrospectively identified by the laboratory information system (LIS). Additionally, if a cerebrospinal fluid (CSF) sample was registered in the LIS within ±3 days of the time of serum sampling, as an indication of suspected CNS infection, the patient was included for medical record and laboratory analysis review.

For 325 patients, the following data were retrieved from the LIS: age at admission, gender, hospital admission, sampling date of the CSF and sampling dates with associated test results for the TICK-BORNE ENCEPHALITIS VIRCLIA IgM/IgG MONOTEST (Vircell, Santa Fe, Granada, Spain) and the ReaScan TBE IgM rapid test (Reagena Oy Ltd., Toivala, Finland). In addition, relevant microbiological diagnostic test results from the same admission were registered.

From the medical records, the following supplementary data were extracted: clinical presentation (meningitis, encephalitis, myelitis), CSF cell count, any TBE notification, previous TBE vaccination (including date and doses, if available), inclusion in the NOTES or prescription of intravenous immunoglobulin (IVIG) therapy.

The presence of a TBE notification in the patients’ medical record defined a TBE case. Thus, the TBE diagnosis was not assessed retrospectively. Furthermore, a TBE VBI case was defined by the presence of a TBE notification in a patient who had obtained at least two vaccine doses and by the time of the last dose according to the definition by Hansson et al. [14].

### 2.2. Laboratory Diagnostics

#### 2.2.1. Sample Material

For validation of the laboratory diagnosis, all reactive VirClia IgM sera in the study period, regardless of VirClia IgG result, were confirmed by the ReaScan IgM. For patients admitted to Vestfold Hospital with suspected CNS infection, VirClia IgM/IgG assays were routinely performed, and for the majority, they were performed in parallel with ReaScan IgM. However, for patients admitted to Telemark Hospital, VirClia IgM/IgG were mainly performed as confirmatory tests after obtaining a reactive ReaScan IgM at site. Thus, negative ReaScan IgM serum samples were only referred for the VirClia IgM/IgG in selected cases, e.g., CNS infection of unknown etiology, or if the ReaScan IgM was not available at the site.

#### 2.2.2. Inclusion of Test Results

In order to compare the two tests retrospectively, only test results obtained by the VirClia IgM and ReaScan IgM from the same serum sample, or from one obtained within ±1 day, were included for statistical analysis. For 20 patients with only a reactive VirClia IgM result available in the LIS, the ReaScan IgM was performed retrospectively from frozen sera in May 2024, while no VirClia IgM/IgG tests were performed retrospectively. All test results were interpreted according to the manufacturer’s instructions without using gray zones developed in-house.

#### 2.2.3. Classification of Test Results

Positive and equivocal test results were interpreted as true positive or false reactive based on the presence or absence of a TBE notification, respectively. The laboratory criteria for a TBE notification were the detection of specific antibodies in serum and/or the CSF or the detection of TBEV RNA in the CSF [7]. The diagnostic criteria for a laboratory-confirmed case were (1) clinical findings consistent with CNS inflammation, (2) pleocytosis (>5 × 10^9^ leukocytes/L) in the CSF and (3) at least one of four laboratory criteria (i.e., detection of specific IgM and IgG antibodies in serum, detection of specific IgM antibodies in the CSF, seroconversion or four-fold increase in IgG in paired sera or detection of TBEV RNA in a clinical specimen) [8]. The Norwegian Institute of Public Health (NIPH) is the reference laboratory and performs an in-house TBEV RNA RT- PCR in suspected cases [15].

### 2.3. Statistical Analysis

Sensitivity, specificity, predictive values and ROC curves were calculated with 95% confidence intervals (CIs) using SPSS (IBM SPSS Statistics for Windows, version 25.0, Armonk, NY, USA: IBM Corp.).

## 3. Results

### 3.1. TBE Cases

A total of 325 patients fulfilled the inclusion criteria, with 85 TBE cases identified, of whom 60% were included in the NOTES. Approximately 65% of the TBE cases were male, with an average age of 52 years (Table 1). The most common clinical presentation was encephalitis with median pleocytosis of 72 × 10^9^ cells/L. Four cases of TBE VBIs were identified, and only two were adequately vaccinated according to the current Norwegian guidelines. Furthermore, one case had received only one vaccine dose, while another had not received booster doses as recommended.

### 3.2. Assay Sensitivity

The VirClia IgM and ReaScan IgM were positive in the initial serum sample in 75 and 80 of the 85 TBE cases, respectively. The estimated sensitivity and positive predictive value (PPV) for the VirClia IgM were 88.2% (95% CI, 79.4–94.2), whereas an estimated sensitivity of 94.1% (95% CI, 86.8–98.1) and PPV of 95.2% (95% CI, 88.3–98.7) were found for the ReaScan IgM. Thus, there were no statistically significant differences regarding the sensitivity and PPV. Due to the different test algorithms, test characteristics for the VirClia IgM were calculated for all patients and for patients admitted to Vestfold Hospital only. However, no statistically significant differences were found (Table 2).

The diagnostic efficiency of the VirClia IgM was analyzed and the area below the ROC curve was 0.968 (95% CI, 0.944–0.992) (Figure 1).

Seventy TBE patients (82%) had positive VirClia IgM and IgG concurrently at admission; five patients had isolated positive VirClia IgM, whereas six patients solely had positive VirClia IgG in serum at admission (Appendix A).

Ten false-negative VirClia IgM cases were identified, and four were categorized as VBIs, of which two were also negative in the ReaScan IgM, and the diagnosis was confirmed by the presence of IgM and IgG in a follow-up sample or by a significant rise in the TBE IgG titer (Table 3). In the two remaining VBI cases, the ReaScan IgM was positive in both the serum and the CSF sample obtained at admission. All VBI cases had positive VirClia IgG in the first serum sample and were above fifty years of age.

The remaining six false-negative VirClia IgM cases were confirmed by the combination of positive ReaScan IgM and VirClia IgG in serum at admission (*n* = 1), positive VirClia IgM and IgG in a follow-up serum sample (*n* = 3) or the detection of TBEV RNA by RT-PCR in the CSF, performed at the NIPH (*n* = 1). Only one patient had a positive ReaScan IgM in serum as the solitary laboratory finding.

### 3.3. Assay Specificity

The estimated specificity and negative predictive value of the VirClia IgM were 95.8% (95% CI, 92.5–98.0) (Table 2). Due to a high number of missing ReaScan IgM results among non-TBE cases (*n* = 210), the calculation of negative predictive value and specificity was not performed for this assay.

Ten cases of false-reactive VirClia IgM were identified, of which three were equivocal and seven were positive test results, according to the manufacturer’s instructions for interpretation (Table 4). Four were concurrently reactive in the ReaScan IgM at admission, and all of these, except for one that was missing, were negative in both the VirClia IgM and VirClia IgG in the follow-up serum sample. Recent TBE immunization and IVIG therapy (during the last 30 days) were plausible causes identified in one case each. On two occasions, the VirClia IgM remained reactive in the follow-up serum sample, without a plausible explanation identified.

## 4. Discussion

Bearing in mind that the detection of IgM alone cannot confirm a TBE diagnosis, VirClia IgM correctly identified 75 of the 85 TBE cases, with an estimated sensitivity and PPV of 88.2%. We are not aware of any previous publications with which to compare this finding, which is significantly lower than the 98% (95% CI, 91–100) provided by the manufacturer [16]. The ReaScan IgM had a sensitivity of 94.1%, but the difference in sensitivity between the tests was not statistically significant. Five cases had no detectable TBE IgM when hospitalized, which is rare according to previous studies [1,17]. However, this result correlates with the findings of the NOTES, where 3.9% (6/153) of the patients had no detectable TBE IgM at admission [4]. This could partly be due to the unusually high prevalence of a monophasic course in the NOTES cohort (46%), which may have affected the dynamic process of antibody production and thus the test results. Another explanation could be the time of sampling in relation to the onset of CNS symptoms. However, among the ten TBE cases with negative VirClia IgM, five had a history of previous TBE vaccination, presenting with isolated positive VirClia IgG in serum at admission. A physiologically delayed TBE IgM production is well described in VBI case reports [8,12,18]. Thus, the number of previously vaccinated patients among the TBE cases will have a major impact on the estimated assay sensitivity.

During the study period, 117 domestic TBE cases were notified in Vestfold and Telemark counties [7]. Thus, this TBE cohort constitutes approximately 73% of all the notified cases. Unreported TBE cases may occur, especially in children with vague symptoms [19]. However, an electronic TBE notification is generated in the case of laboratory confirmation. Thus, there is very little likelihood that a clinical case will go unreported. Predominantly, the TBE cases in our study were between the ages of 40 and 64, with a majority of male patients, corroborating the findings from elsewhere in Europe [2]. The unusually high occurrence of encephalitis, of almost 70%, is in accordance with the results published in the NOTES [4].

Two patients had no detectable TBE antibodies initially. One immunocompromised participant was diagnosed through the detection of TBEV RNA in the CSF [4]. This case illustrates the importance of TBEV RNA detection in serum or the CSF in patients with typical clinical illness but without detectable antibodies [20,21].

The VirClia IgM had a very high specificity of 95.8% in our study, which is only slightly lower than the 99% which is reported by the assay manufacturer [16]. In comparison, the specificity of the ReaScan IgM had previously been estimated to be 97.7% in a multi-laboratory evaluation when compared to different ELISAs and EIA assays [13]. The different test algorithms for the VirClia IgM at the two hospitals might have had an impact on the calculated specificity. However, the difference in specificity was not statistically significant when patients from Telemark Hospital were excluded from the analysis.

Ten non-TBE patients had a false-reactive VirClia IgM serum sample at admission, four with concurrent reactivity in the ReaScan IgM, of which one case was due to an acute Epstein–Barr virus infection, one had a recent TBE immunization (first dose), one was a case of human herpesvirus 6 meningitis and, finally, one was a child with aseptic meningitis without any identified causative agent. The six isolated false-reactive VirClia IgM cases included one patient with neuroborreliosis and another patient suffering from Cryptococcus neoformans infection who had been given IVIG therapy during the last month upon admission. However, whether the latter reactivity was due to passive antibody transfer or cross-reactivity remains uncertain. For the remaining four cases, no plausible explanation for the false-reactive VirClia IgM could be found. In such cases, a follow-up serum sample is needed to detect an IgG seroconversion or rule out TBE infection.

A significant weakness of this study is the retrospective inclusion. The TBE diagnoses were not assessed retrospectively but were based on case notifications. Although we found no unreported TBE cases among patients with a reactive VirClia IgM, a prospective inclusion would have been optimal to assess false-negative cases. However, this study benefits from the contemporary prospective inclusion of TBE patients to the NOTES, in which 60% of the cases in our study also participated. The prospective inclusion most likely increased the clinicians’ awareness of TBE as a differential diagnosis, ensuring that necessary clinical specimens were obtained for the laboratory confirmation, especially in patients who are more challenging to diagnose, i.e., immunocompromised and previously vaccinated patients. The participants who were not included in the NOTES were mainly patients admitted after the inclusion for that study was completed or children (<16 years) who were not eligible to participate in that study.

For the majority of the TBE cases, there were no information regarding previous TBE vaccination. Thus, potentially, some VBI cases might have been undiscovered. Only two VBI cases in our study were adequately vaccinated according to the current guidelines: a man in his fifties with three doses and a woman in her sixties with four doses as part of their primary immunization. Neither was immunosuppressed. The main factors for the immunological response to TBE vaccination are age and the number of vaccine doses [22]. The current Norwegian guidelines takes this into account and recommend four, instead of three, doses in the primary immunization in individuals at the age of 60 and above [23]. TBE vaccination is still uncommon in the general population in endemic regions in Norway, as it is not included in any vaccination program. Data from the Norwegian vaccine register revealed that the number of individuals receiving the TBE vaccine nearly tripled in 2023 compared to 2021, most likely due to higher awareness among the target groups [6].

Interference in TBEV-specific ELISA assays caused by flavivirus cross-reactive antibodies is well documented [8,24]. Globally, the most important human pathogen flaviviruses, e.g., West Nile virus, Japanese encephalitis virus and dengue virus, are not endemic in Norway and hence are of little concern. However, infections can be acquired during travels, and occasionally travelers get vaccinated against the latter two. Approximately 50 cases of dengue fever have been notified in Norway yearly since 2014, in contrary to the nil and four cases ever reported of West Nile fever and Japanese encephalitis, respectively [6]. Nevertheless, in travelers returning with fever, laboratories must be aware of potentially cross-reactive antibodies interfering with TBEV IgG assays. Estimating the specificity of the VirClia IgG was not an objective in this study but is highly warranted.

At present, only two flaviviruses are circulating in Norway. Louping ill virus (LIV), which is genetically closely related to TBEV, is also transmitted by *I. ricinus* and may cause encephalomyelitis, mainly in sheep [25]. Although the last reported Norwegian animal case of LIV was back in 1991, the virus still co-circulates with TBEV in southern Norway [26]. Human LIV infections, mainly laboratory-acquired or in other exposed professionals, have been reported in the United Kingdom, but to the best of our knowledge, no cases of human LIV infections have ever been reported in Norway [25,27]. Interestingly, vaccination against TBE seems to be protective against LIV due to cross-neutralizing antibodies [24]. Hence, cross-reactivity due to LIV in TBEV IgG assays may potentially occur.

## 5. Conclusions

To the best of our knowledge, this is the first clinical evaluation of the VirClia IgM, which presents good diagnostic efficiency and was non-inferior to the ReaScan IgM for the diagnosis of TBE in this large cohort. However, an isolated reactive VirClia IgM must be interpreted with caution, since false-reactive results do occur.

## Figures and Tables

**Figure 1 viruses-16-01505-f001:**
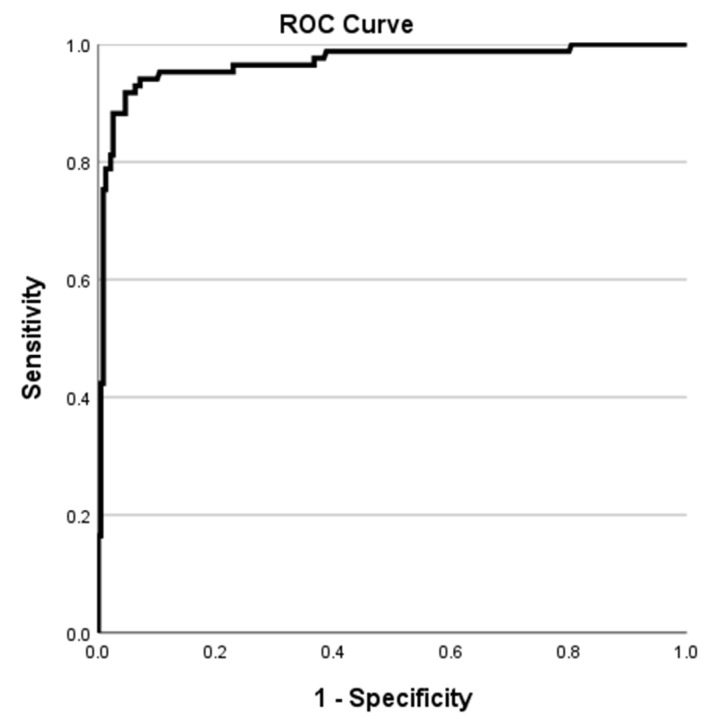
Diagnostic efficiency curve (ROC curve) for the VirClia IgM. The area below the curve is 0.968 (95% CI, 0.944–0.992).

**Table 1 viruses-16-01505-t001:** Patient characteristics and CSF cell count for 85 TBE cases.

		*n*	%
Age at admission, mean (range)		52 (5–87)	
Sex	Male	55	64.7
Hospital admission	Vestfold	34	40.0
CSF cell count (leukocytes × 10^9^/L), median (IQR ^1^)		72 (33–174)	
Previous TBE immunization			
Yes		6	7.1
No		18	21.2
Missing		61	71.8
Clinical presentation			
Meningitis		26	30.6
Encephalitis		58	68.2
Myelitis		1	1.2

^1^ IQR, interquartile range.

**Table 2 viruses-16-01505-t002:** Sensitivity, specificity, positive predictive value (PPV) and negative predictive value (NPV) of VirClia IgM.

Patient Population	VirClia IgM		*n*	Sensitivity	Specificity	PPV	NPV
	Positive	Negative		(95% CI) ^1^	(95% CI) ^1^	(95% CI) ^1^	(95% CI) ^1^
TBE patients, all	75	10	85	88.2%	95.8%	88.2%	95.8%
Non-TBE patients, all	10	230	240	(79.4–94.2)	(92.5–98.0)	(79.4–94.2)	(92.5–98.0)
TBE patients, only Vestfold	31	3	34	91.2%	96.2%	79.5%	98.6%
Non-TBE patients, only Vestfold	8	205	213	(76.3–98.1)	(92.7–98.4)	(63.5–90.7)	(95.8–99.7)

^1^ Calculated by Clopper–Pearson exact method.

**Table 3 viruses-16-01505-t003:** Test results and TBE vaccination status for TBE cases with false-negative VirClia IgM serum sample at admission.

Age	Test Result—First Serum Sample			Test Result—Follow-Up Serum Sample			TBE Vaccination		
	VirClia IgM	VirClia IgG	ReaScan IgM	VirClia IgM	VirClia IgG	Days ^a^	Doses	Year ^b^	VBI ^c^
54	Negative	Positive	Negative	Negative	Positive ^1^	30	3	2021	Yes
77	Negative	Positive	Negative ^2^	Positive ^3^	Positive	13	3	2022	Yes
41	Negative	Negative	Negative	Positive	Positive	9	-	-	-
49	Negative	Negative	Positive	Missing	Missing	-	-	-	-
62	Negative	Positive	Negative	Positive ^4^	Positive	9	3	2012	No
52	Negative	Positive	Positive	Missing	Missing	-	-	-	-
64	Negative	Positive	Positive ^5^	Missing	Missing	-	2	2023	Yes
69	Negative	Positive	Positive ^5^	Missing	Missing	-	4	2021	Yes
67	Negative ^6^	Negative	Negative	Positive	Negative	17	-	-	-
38	Negative	Negative	Positive	Positive	Positiv	10	-	-	-

^a^ Days between the first serum sample at admission and follow-up serum ample. ^b^ Year in which the last vaccine dose was received. ^c^ Defined as TBE after ≥2 vaccine doses and if the last dose was received within >4 weeks and <12 months (if 2 doses), <36 months (if 3 doses) and <60 months (if ≥4 doses). ^1^ Significant rise in SERION ELISA classic FSME/TBE Virus IgG titer, performed at the NIPH. ^2^ Negative ReaScan IgM in CSF sample obtained on the same day. ^3^ Negative VirClia IgM in a follow-up serum sample obtained 5 days after the first serum sample. ^4^ Negative TBEV RNA RT-PCR in serum, performed at the NIPH. ^5^ Positive ReaScan IgM in CSF sample obtained on the same day. ^6^ Positive TBE RNA RT-PCT in CSF, performed at the NIPH.

**Table 4 viruses-16-01505-t004:** Test results for non-TBE cases with false-reactive VirClia IgM serum samples at admission.

Age	Test Result—First Serum Sample			CSF Cell Count ^a^	Test Result—Follow-Up Serum Sample		
	VirClia IgM	VirClia IgG	ReaScan IgM		VirClia IgM	VirClia IgG	Days ^b^
26	Positive	Negative	Negative	11–49	Negative	Negative	25
33	Positive	Negative	Negative	<6	Positive	Negative	22
72	Positive	Negative	Negative	<6	Missing	Missing	-
38	Positive	Negative	Negative	<6	Positive ^1^	Negative	4
4	Equivocal ^2^	Negative	Positive	11–49	Negative	Negative	10
18	Positive	Negative	Negative	11–49	Missing	Missing	-
16	Equivocal	Negative	Positive	100–199	Missing	Missing	-
56	Equivocal ^3^	Negative	Negative	100–199	Missing	Missing	-
78	Positive ^4^	Negative	Equivocal	<6	Negative	Negative	90
23	Positive	Negative	Equivocal	11–49	Negative	Negative	200

^a^ Leucocytes × 10^9^/L. ^b^ Days between the first serum sample at admission and follow-up serum. ^1^ Negative ReaScan IgM sample. ^2^ Negative TBEV RNA RT-PCR in serum, performed at the NIPH. ^3^ IVIG therapy during the last 30 days. ^4^ Recent TBE immunization (1 dose) according to the medical record.

## Data Availability

The anonymous case data presented in this study are available on request from the corresponding author.

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
