# Peer review of "Clinical Evaluation of the VirClia IgM/IgG Chemiluminescence Tests for the Diagnosis of Tick-Borne Encephalitis in an Endemic Part of Norway"

_viruses, 2024, doi:10.3390/v16091505_

Round 1
Reviewer 1 Report
Comments and Suggestions for Authors
Comments to Clinical evaluation of the chemiluminescence tests VirClia IgM/IgG for the diagnosis of tick-borne encephalitis in an endemic part of Norway. viruses-3219041
General comments
This paper provides a clinical /retrospective evaluation of a serological test for TBE-virus infections in a Norwegian population. The population is defined by clear inclusion criteria, and other methodological standards and statistical evaluation are appropriately chosen. The result section is fine and short, and provides some interesting details, while the discussion perhaps is a bit too short. Overall a nice and tidy paper for a neat study.
Major comments.
1. Section 2.2 suggests that the authors extracted information on serology from patient files and that ReaScan results already were available from some patients (Telemeark). If I understand correctly then you should be able to assess the reproducibility of the ReaScan outcome in patients from Telemark, which could be helpful in the discussion. Please add some sample numbers to the 2.2. section as it would be easier to understand what you did. How much info was extracted and how many serological assays did you perform yourself?
2. You will need to comment a bit more on your chosen reference i.e. that only notified cases we “true” TBE cases, while unnotified cases were not. Reporting systems are rarely highly reliable, and it is particularly difficult to discuss “false negative” findings – as someone might simply have forgotten to do the paperwork. Pls. add a discussion of these issues.
3. I find it a bit odd that issues relating to Louping Ill are not discussed, just as I would expect some sort of mentioning of other viruses that have cross-reactivity to TBEV - West Nile fever and Japanese encephalitis.
I would suspect that your inclusion criteria in most cases would remove potential louping ill infections – but please investigate the literature and assess whether there could be exceptions. Pls also make appropriate mention of the Japanese encephalitis and West Nile fever in relation to their current distribution.
Past findings of Louping ill e.g., Ytrehus, B., Vainio, K., Dudman, S. G., Gilray, J., & Willoughby, K. (2013). Tick-borne encephalitis virus and louping-ill virus may co-circulate in Southern Norway. Vector-Borne and Zoonotic Diseases, 13(10), 762-768.
See map of Louping ill distribution in Norway in Ytrehus, B., Rocchi, M., Brandsegg, H., Turnbull, D., Miller, A., Pedersen, H. C., ... & Nilsen, E. B. (2021). Louping-ill virus serosurvey of Willow Ptarmigan (Lagopus lagopus lagopus) in Norway. The Journal of Wildlife Diseases, 57(2), 282-291.
Minor comments
I expect that there at some point someone must have compared the serological methods with the neutralization test. Pls., include a mention of these such that we can better appreciate the specificity of the method. And do these records include information on the cross-reactivity with other flaviviruses?
Result section – I think you need to shift some of the tables down to the discussion section – because the relatively short result section is difficult to read when it is broken up into many very short paragraphs.
Tables. Table 3 – is broken by a page shift – pls. rearrange the text.
Figure- No comment.
Author Response
Response to Reviewer 1
Comment 1: Section 2.2 suggests that the author’s extracted information on serology from patient files and that ReaScan results already were available from some patients (Telemark). If I understand correctly then you should be able to assess the reproducibility of the ReaScan outcome in patients from Telemark, which could be helpful in the discussion. Please add some sample numbers to the 2.2. section as it would be easier to understand what you did. How much info was extracted and how many serological assays did you perform yourself?
Response 1: Thank you for pointing out that this section was a bit unclear. We totally agree and have therefore divided the section “2.2. Laboratory diagnostics” into 3 parts (“2.2.1. Sample material” page 2, line 84, “2.2.2. Inclusion of test results” page 2 line 94 and “2.2.3. Classification of test results” page 3 line 103). Regarding the serological assays, we have specified that mainly reactive ReaScan IgM serum samples were referred from Telemark Hospital, while negative ReaScan IgM serum samples were only referred in selected cases, by adding a new sentence in “2.2.1 Sample material” page 2 line 90; “Thus, negative ReaScan IgM serum samples were only referred for VirClia IgM/IgG in selected cases, e.g. CNS infection of unknown etiology, or if ReaScan IgM was not available at site.” Thus, potentially the majority of negative ReaScan IgM serum samples were not analyzed by the VirClia assay. We have also specified on page 3 line 97 that 20 ReaScan IgM samples were performed retrospectively and on line 100 that none VirClia IgM tests were performed retrospectively. We have also added new information in section 2.2.3 page 3 line 100/101 marked in red; “All test results were interpreted according to the manufacturer’s instructions without using in-house developed grey zones.” Finally, we have specified in section 2.1. , page 2 line 66 that the patient was included for medical record and laboratory analysis review.
Comment 2: You will need to comment a bit more on your chosen reference i.e. that only notified cases we “true” TBE cases, while unnotified cases were not. Reporting systems are rarely highly reliable, and it is particularly difficult to discuss “false negative” findings – as someone might simply have forgotten to do the paperwork. Pls. add a discussion of these issues.
Response 2: Thank you for pointing this out. We have revised the discussion to emphasize this point by 1) adding the following sentences in section 4, page 6 line 196; “Unreported TBE cases may occur, especially in children with vague symptoms [9]. However, an electronic TBE notification is generated in the case of a laboratory confirmation. Thus, there is very little likelihood that a clinical case will go unreported.” 2) Adding the following sentences in the same section page 7 line 225-228; “The TBE diagnosis were not assessed retrospectively, but were based on case notifications. Although we found none unreported TBE cases among patients with a reactive VirClia IgM, a prospective inclusion would be optimal to assess false negative cases.” And 3) specified that inclusion to the simultaneous NOTES study was prospective on line 229 and 230.
Comment 3: I find it a bit odd that issues relating to Louping Ill are not discussed, just as I would expect some sort of mentioning of other viruses that have cross-reactivity to TBEV - West Nile fever and Japanese encephalitis.
I would suspect that your inclusion criteria in most cases would remove potential louping ill infections – but please investigate the literature and assess whether there could be exceptions. Pls also make appropriate mention of the Japanese encephalitis and West Nile fever in relation to their current distribution.
Past findings of Louping ill e.g., Ytrehus, B., Vainio, K., Dudman, S. G., Gilray, J., & Willoughby, K. (2013). Tick-borne encephalitis virus and louping-ill virus may co-circulate in Southern Norway. Vector-Borne and Zoonotic Diseases, 13(10), 762-768.
See map of Louping ill distribution in Norway in Ytrehus, B., Rocchi, M., Brandsegg, H., Turnbull, D., Miller, A., Pedersen, H. C., ... & Nilsen, E. B. (2021). Louping-ill virus serosurvey of Willow Ptarmigan (Lagopus lagopus lagopus) in Norway. The Journal of Wildlife Diseases, 57(2), 282-291.
Response 3: Agree. We have therefore replaced the last paragraph in the discussion, “Infections with other flaviviruses are not a major concern in Norway, and so far, a neutralization assay is not available. Nevertheless, in travelers returning with fever, laboratories must be aware that cross-reactive antibodies e.g. Dengue virus IgG might interfere with the assays [6]”, with an elaborated paragraph (page 8 line 249- page 9 line 267):
“Interference in TBEV specific ELISA assays caused by flavivirus cross-reactive anti-bodies is well documented [8, 24]. Globally, the most important human pathogen flaviviruses, e.g. West Nile virus, Japanese encephalitis virus and dengue virus, are not endemic in Norway, hence of little concern. However, infections can be acquired during travels and occasionally travelers get vaccinated against the latter two. Approximately, 50 cases of dengue fever have been notified in Norway yearly since 2014, in contrary to the nil and four cases ever reported of West Nile fever and Japanese encephalitis, respectively [6]. Nevertheless, in travelers returning with fever, laboratories must be aware of potentially cross-reactive antibodies interfering with TBEV IgG assays. Estimating the specificity of the VirClia IgG was not an objective in this study, but is highly warranted.
At present, only two flaviviruses are circulating in Norway. Louping ill virus (LIV), which is genetically closely related to TBEV, is also transmitted by I. ricinus and may cause encephalomyelitis mainly in sheep [25]. Although the last reported Norwegian animal case of LIV was back in 1991, the virus still co-circulates with TBEV in southern Norway [26]. Human LIV infections are reported from the United Kingdom, mainly laboratory-acquired or in other exposed professionals, but to the best of our knowledge, no cases of human LIV infections have ever been reported in Norway [25, 27]. Interestingly, vaccination against TBE seems protective against LIV due to cross-neutralizing antibodies [24]. Hence cross-reactivity due to LIV in TBEV IgG assays may potentially occur.”
Comment 4: I expect that there at some point someone must have compared the serological methods with the neutralization test. Pls., include a mention of these such that we can better appreciate the specificity of the method. And do these records include information on the cross-reactivity with other flaviviruses?
Response 4: We totally agree with this expectation. However, to the best of our knowledge, there are no publications comparing ReaScan IgM or VirClia IgM with a neutralization assay. In the multi-laboratory evaluation of ReaScan TBE IgM rapid test, the specificity was assessed by the commercial Enzygnost Anti-TBE virus IgG test and an in-house EIA, and we find no other publications on PubMed or provided by the manufacturer. In the discussion page 7 line 210 we have added “when compared to different ELISA and EIA assays” to the end of the sentence “In comparison, the specificity for ReaScan IgM has previously been estimated to 97.7 % in a multi-laboratory evaluation”. Regarding the VirClia IgM, the manufacturer informs in the packet-insert that the sensitivity and specificity were assessed by the use of a commercial ELISA kit.
Comment result section: I think you need to shift some of the tables down to the discussion section – because the relatively short result section is difficult to read when it is broken up into many very short paragraphs.
Response result section: We agree that the result section is difficult to read. However, we would like to remove the former “Table 3” from the manuscript and provide it as a supplementary table, “Table S1”. In addition, we have placed Table 3 and 4 to the end of the result section.
Comment tables: Table 3 – is broken by a page shift – pls. rearrange the text.
Response tables: Agree. We have modified “Table 1” so that it fits into page 3. The main change is that the CSF cell count now is displayed on only one line with the median value and interquartile range. We have therefore replaced the former “low-grade pleocytosis (6-49 x 109 cells/L)” in section 3.1, page 3 line 123 with median pleocytosis of 72 x 109 cells/L.

Reviewer 2 Report
Comments and Suggestions for Authors
The manuscript submitted to me for review appears well structured and certainly original.
I would only have a few observations/suggestions to make:
Abstract: structured and complete.
Introduction: page 1, line 42: it would be appropriate to specify what change in land and human behaviour means.
Mat & Meth: it would be appropriate to also report the number of subjects involved (which appears at the beginning of the results) in this section.
Results: well presented. I appreciate the use of ROC analysis.
Discussion: Consistent with the results as well as the conclusions.
References: exhaustive.
Author Response
Response to Reviewer 2
Comment 1: Introduction: page 1, line 42: it would be appropriate to specify what change in land and human behaviour means.
Resonse 2: Thank you for pointing this out. We have revised the end of the sentence (marked in red) in the introduction page 1 line 42 to “Until 2015, the incidence of TBE fluctuated, but thereafter Norway saw a gradual increase in TBE cases, likely due to a combination of factors increasing the I. ricinus distribution and human exposure. We have also added a new sentence p 1, line 42-44, included a new reference (number 5); Climate changes, forest regrowth due to e.g. decreasing number of farms and people spending more time outdoors, especially during the COVID-19 pandemic, are probably all contributing factors [5, 6].
- Jore, Solveig.; Vanwambeke, S.O.; Viljugrein, H.; Isaksen, K.; Kristoffersen, A.B. Woldehiwer, Z.; et al. "Climate and environmental change drives Ixodes ricinus geographical expansion at the northern range margin." Parasites & vectors 2014, 7, 1-14. https://doi.org/10.1186/1756-3305-7-11
Comment 2: Mat & Meth: it would be appropriate to also report the number of subjects involved (which appears at the beginning of the results) in this section.
Response 2: We agree and have therefore specified in section 2.1, page 2 line 68 that for 325 patients data were retrieved from the laboratory information system, to make it even clearer. We have also pointed out that it was initially 508 patients who had a VirClia IgM result, of whom 325 fulfilled the criteria for medical record review, by adding 508 to the first sentence in the section 2.1 page 2 line 62.